# Investigating the Accuracy of Ultrasound Imaging in Measuring Fetal Weight in Comparison with the Actual Postpartum Weight

**DOI:** 10.3390/pediatric17040070

**Published:** 2025-06-24

**Authors:** Sultan Abdulwadoud Alshoabi, Abdulhadi M. Tarshun, Ziyad O. Alnoman, Fahad H. Aljohani, Fadwa M. Alahmadi, Awatif M. Omer, Osamah M. Abdulaal, Awadia Gareeballah, Abdulaziz A. Qurashi, Fahad H. Alhazmi, Kamal D. Alsultan, Moawia Gameraddin

**Affiliations:** 1Department of Diagnostic Radiology, College of Applied Medical Sciences, Taibah University, Al-Madinah Al-Munawwarah 42353, Saudi Arabia; 2Department of Gynecology and Obstetrics, Maternity and Children Hospital, King Salman bin Abdulaziz Medical City, Al-Madinah Al-Munawwarah 42319, Saudi Arabia

**Keywords:** antenatal ultrasonography, estimated fetal weight (EFW), birth weight (BW), Hadlock A formula, head circumference (HC), abdominal circumference (AC), femur length (FL)

## Abstract

Background: Antenatal ultrasonography measurements of the estimated fetal weight (EFW) are a critical point in the decision-making process of obstetric planning and management to preserve the safety of both the newborn and the mother. This study aims to investigate the accuracy of ultrasonography to measure the EFW in comparison with the actual birth weight (BW) measured immediately after delivery. Methods: In this retrospective study, electronic records of 270 newborns who fulfilled the inclusion criteria were retrieved. A structured data sheet was used to collect the EFW, calculated by the Hadlock A formula using real-time ultrasound imaging on the day of delivery or the day before, and the actual BW immediately after delivery. Results: Out of 270 fetuses, 53.7% (145) were female, and 46.3% (125) were male. The mean BW was 2918.1 ± 652.81 g (range: 880 to 5100). The mean EFW was 3271.55 ± 691.47 g (range: 951 to 4942). The mean gestational age was 38 ± 2.48 weeks (range: 29 to 42). Spearman’s rho correlation test revealed strong compatibility between EFW and BW (r = 0.82, *p* < 0.001). Linear regression analysis showed a strong correlation between EFW and BW (R = 0.875, R^2^ = 0.766, and *p* < 0.001). The cross-tabulation test showed 86.8%, 78.4%, and 26.9% compatibility between measurements of EFW and the true BW in group-1 (<2500 g), group-2 (2500–4000 g), and group-3 (>4000 g) fetuses (*p*< 0.001). Conclusions: EFW using ultrasonography yields high compatibility with the actual BW. Despite the slight overestimation, ultrasonography provides high clinical value in obstetric assessment and subsequent management.

## 1. Introduction

Ultrasonography is an imaging modality that uses a high frequency sound beam greater than human hearing limits (20 KHz) to visualize the anatomical structures inside the human body [1]. Antenatal ultrasonography measurements of the estimated fetal weight (EFW) are a critical point in the decision-making process of obstetric planning and management to preserve the safety of both the newborn and the mother. Ultrasound operators usually use three measurements input into an algorithm designed by Hadlock et al. to calculate the fetal weight during pregnancy. Fetal measurements taken by an experienced ultrasound operator include biparietal diameter (BPD), head circumference (HC), abdominal circumference (AC), and femur length (FL) [2].

Measuring the EFW is a vital and universal part of antenatal care for managing growth monitoring, high-risk pregnancies, and labor and delivery. Both high and low fetal weights are associated with an increased risk of newborn complications during delivery and postpartum. Low birth weight may affect perinatal morbidity and mortality due to low preterm delivery and intrauterine fetal death. High fetal weight may affect shoulder dystocia, brachial plexus injury, bone injuries, and intrapartum asphyxia. Maternal risks include pelvic floor injuries and postpartum hemorrhage [3]. EFW can be very influential in determining the mode and timing of labor [4]. EFW calculated using ultrasonography is more accurate than fetal weight calculated by clinical methods such as Johnson’s method and Insler and Bernstein’s formula, with ultrasonography having a focus on studying fetal viability, organs, and the surrounding environment [5]. Ultrasonography is a non-invasive, radiation-free, widely available, and affordable imaging modality that uses color and pulsed Doppler to study human organs and blood perfusion [6].

Due to the importance of measuring the weight of fetuses and the role of ultrasonography, this study aims to investigate the accuracy of ultrasonography in measuring the EFW by comparing the mean EFW measured on the day of delivery or the day before with the actual birth weight (BW) measured immediately after delivery. In addition, this study will measure the average BWs of fetuses, compare the average weights of male and female fetuses, and correlate fetal weight to the body mass index (BMI), height, and weight of the mother. In our region, we did not find previous studies covering this significant topic. This is the first study exploring this critical topic in our region. We hypothesized that ultrasonography, when performed by an experienced operator, is an accurate imaging method in measuring weight of the fetus.

## 2. Patients and Methods

### 2.1. Study Design

Institutional ethical approval (IRB23-029) was granted for this study. This was a retrospective cross-sectional study, with data collection involving the electronic records of 270 newborns from the Picture Archiving and Communication System (PACS) system of the Maternity and Children Hospital, Almadinah Almunawwarah, Kingdom of Saudi Arabia. A structured data sheet was used to collect the mother’s age, weight, and height, as well as the EFW, calculated by ultrasound imaging on the day of delivery or the day before, and the actual BW after delivery. This study involved all singleton pregnancies with a healthy newborn and available ultrasonography performed in the last 48 h before delivery, regardless of the value of the EFW. Exclusion criteria were as follows: (1) newborns with no available ultrasound imaging in the last two days of pregnancy and (2) newborns with any chromosomal or congenital abnormalities.

### 2.2. Procedure for EFW

Ultrasonography EFW was obtained for all women by the same three radiographers, each with more than five years of experience in gynecology and obstetric ultrasonography. They performed real-time examinations using a 3.5 MHz electronically focused transducer (Voluson E10 BT17, Austria) and the standard Hadlock reference tables, which use biparietal diameter (BPD), occipitofrontal diameter (OFD), head circumference (HC), abdominal circumference (AC), and femur length (FL) to calculate fetal weight (Figure 1).

All fetuses’ EFWs were calculated using the Hadlock formula, which is documented by Engelbrechtsen et al. [7]. The Hadlock formula, based on measurements of HC, AC, and FL, was used during the ultrasound scan [8]. The fetal AC was calculated by measuring the abdominal transverse diameter (ATD) and antero-posterior diameter (APD) in a cross-sectional scan when the umbilical vein and stomach bubble were visible in the anterior third at the portal sinus level, using the following formula [9]:[AC = π × (ATD + APD)/2]

HC was calculated using the following formula [9]:[HC = π × (OFD + BPD)/2].

According to Kong et al., the Hadlock 1 formula is closest to being the most accurate, with the lowest error [10]. However, Mattsson et al. reported that Hadlock 2, which uses three parameters (BPD, AC, and FL), gave the highest correlation coefficient (0.91) with actual BW [11]. The immediately calculated postpartum neonatal BW was considered as the gold standard.

### 2.3. Statistical Analysis

The collected data were organized, prepared, and analyzed using Statistical Package for the Social Sciences (SPSS) Statistics version 27 (IBM, Armonk, NY, USA). Descriptive statistics, including means ± standard deviation (SD), were used for continuous parameters. Frequencies with relative percentage (%) were used for categorical variables. The Spearman’s rho correlation test was conducted to assess the compatibility between the EFW and actual BW. Shapiro–Wilk and Kolmogorov–Smirnov tests were conducted and showed a non-normal distribution of the EFW and BW measurements (*p*-value < 0.001). Histogram analysis was also confirmed non-normal distribution A Mann–Whitney U non-parametric statistical test was used to compare the actual BW between male and female newborns. A cross-tabulation test was used to assess the compatibility between the EFW and the actual BW measurements in group-1 (<2500 g), group-2 (2500–4000 g), and group-3 (>4000 g) fetuses. The *p*-value resulting from two-sided statistical tests was assumed to be statistically significant when less than 0.05.

## 3. Results

In total, 270 mothers with singleton pregnancies were involved in this study, with a mean age of 30.65 ± 6.1 years (range: 18 to 45), a mean weight of 71.63 ± 14.24 years (range: 40 to 130), a mean height 157.2 ± 5.8 kg (range: 140 to 180), and a mean BMI of 28.83 ± 5.55 (range: 17 to 53).

Of the 270 fetuses (n = 270), 53.7% (145) were female and 46.3% (125) were male. Those who fulfilled the inclusion criteria underwent ultrasonography measurement of EFW, which was compared with the actual BW. The BW in the study population ranged from 880 to 5100 g, with a mean of 2918.1 ± 652.81 g. The EFW in the study population ranged from 951 to 4942 g, with a mean of 3271.55 ± 691.47 g. The mean gestational age was 38 ± 2.48 weeks, ranging from 29 to 42 weeks. The maximum interval between the EFW ultrasound examination and birth was 48 h.

Our results show wide variation in the fetal weight values measured using ultrasonography (*p* < 0.001) and wide variation between the EFW and BW values measured after delivery (*p* < 0.001) (Table 1).

In addition, Spearman’s rho correlation test revealed strong compatibility between EFW and the actual BW measurements immediately after delivery (r = 0.82, *p* < 0.001).

In addition, simple linear regression analysis showed a strong linear correlation between EFW and actual BW (R = 0.875, R^2^ = 0.766, and *p* < 0.001) (Figure 2).

In the comparison between weight groups, the cross-tabulation test shows 86.8%, 78.4%, and 26.9% compatibility between measurements of EFWs and the actual BW in group-1, group-2, and group-3 fetuses (*p* < 0.001). Specifically, 14.1% of EFWs assigned to group-2 were actually in group-1, and 73.1% of the EFWs assigned to group-3 were actually in group-2. This means that ultrasonography tends to overestimate the EFW, especially in high BW fetuses (Table 2).

The boxplot shows that ultrasonography tends to slightly overestimate fetal weight in the three groups (Figure 3).

A Mann–Whitney U test was conducted to compare actual BW between male and female fetuses. The test showed that the difference between males and females with respect to BW was not statistically significant (U = 7890, *p* = 0.067), with only a small effect size (r = 0.11). The mean, median, and mean rank of BW for males were slightly higher than those for females (Table 3).

In addition, a multivariate linear regression was conducted to show that the mother’s weight, height, and BMI had no statistically significant effect on the prediction of the BW of the fetus (R = 0.169, R^2^ = 0.028). The dependent variable was BW and the predictors were mother’s weight, height, and BMI (Table 4).

## 4. Discussion

Antenatal ultrasonography measurements of the EFW are of paramount importance in the decision-making process of obstetric planning and management. In this study, we investigated the accuracy of ultrasonography in measuring the EFW on the day of delivery or the day before and compared it with the actual BW measured immediately after delivery. We found a strong compatibility between ultrasonography EFW and BW measurements of fetal weight (3271.55 ± 691.47 g vs. 2918.08 ± 652.79 g, r = 0.82, *p* < 0.001). This result is strongly compatible with the results of a previous study by Tawe et al., who reported a strong correlation between ultrasonography EFW and actual BW (r = 0.835) and found the difference was not statistically significant (*p* < 0.001) [12]. Furthermore, the strong compatibility between EFW and BW is also in line with a previous study by Eze et al., who reported that EFW is strongly correlated with actual BW and that ultrasonography is an accurate predictor of low weight and overweight fetuses [13]. Kehl et al. reported that most of the currently used equations for calculating EFW have reached their best accuracy using conventional parameters on two-dimensional ultrasonography, within a range of ±500 g [14]. Moreover, Kang et al. reported that a fetal weight prediction model established by semi-automatic three-dimensional limb volume combined with AC has high accuracy, sensitivity, and specificity with high predictive efficacy for the diagnosis of macrosomia (birth weight > 4000 g) [15].

In a previous study, Milner et al. also reported that the accuracy of ultrasonography EFW measurements is related to the formula used and the number of incorporated biometric parameters, and that the Hadlock A formula remains the most reliable method with the smallest random error [16]. In our study, the EFWs were calculated by using the Hadlock A formula with three parameters. Our results showed that ultrasonography tends to overestimate fetal weight, especially in cases of high BW. These results are in line with a previous study by Milner et al., who reported that fetal weight measured by ultrasonography is commonly overestimated in comparison with the actual weight [16]. Our results are further supported by a study by Dittkrist et al., who reported that the lower accuracy of ultrasonography in measuring EFW is associated with high BW [17]. Furthermore, a previous study by Stubert et al. reported that newborns with low BW were more frequently overestimated, while newborns with high BW were more frequently underestimated [18]. This is compatible with our results regarding low BW fetuses, but contrasts with our findings for high BW fetuses. Our results are supported by another study by Ridha et al., who also reported that the accuracy of EFW overall is good, though it tends to overestimate fetuses at low gestational ages [19].

Our results showed that EFW is significantly affected by the fetal gender (*p* = 0.027, 95% CI 5.238–143.235). Broere-Brown et al. reported a gender difference in fetal growth during the first trimester, with males showing a higher crown–rump length. Males have a lower EFW in the second trimester, but no difference in EFW is observed in the third trimester. They also reported that males have a higher BW than females [20]. Male fetuses are larger than females, with some disparity, as reported by Kiserud et al. in their study that provides the World Health Organization fetal growth charts for EFW and shows variations in different parts of the world [21]. Ultimately, our results show that gestational age has a strong effect on EFW. This intuitive and logical relationship shows that fetal weight increases with GA. This finding is augmented by a previous study by Mongelli et al., who reported that intrauterine fetal growth in the third trimester is quasi-linear [22].

Our study showed that fetal weight is not affected by the mother’s weight (*p* = 0.666, 95% CI −40.7–63.6), height (*p* = 0.964, 95% CI −50.35–52.69), or BMI (*p* = 0.857, 95% CI −140.6–117.1). These results are not compatible with the results of Alfadhli, who reported that maternal obesity may be associated with higher BW [23]. Regarding BMI, a previous study reported that high maternal BMI, whether overweight or obesity, appeared to increase the risk of low BW and preterm birth [24]. In contrast to our results, Zhang et al. reported a strong association between maternal height and fetal growth measures, including birth length and BW. Shorter mothers deliver newborns at an earlier gestational age with lower BW and birth length than taller mothers [25]. Witter et al. reported a significant difference in BW and birth height, with newborns of shorter mothers being shorter than those of taller mothers, especially after 35 weeks of GA [26]. Furthermore, Softa et al. reported a positive association between maternal weight and newborn BW [27]. Ultimately, the relationship between BW and the mother’s weight and height is still ambiguous and requires further studies to clarify.

Limitations: This study is limited by its retrospective nature. The EFW data were collected from the PACS system and calculated by three different ultrasonography operators, taking into account the differences in their skills in measuring the EFW. The relatively small sample size may have affected the statical power of some analyses, such as the incompatibility between our results and previous studies regarding the correlation between fetal weight and the mother’s weight, height, and BMI. Further studies with prospective data collection and EFW determined by more than operator are recommended.

## 5. Conclusions

This study concluded that EFW measured using ultrasonography yields a high compatibility with the actual BW. Despite the slight overestimation of EFW, ultrasonography is accurate in measuring fetal weight and provides high clinical value in obstetric assessment and subsequent management.

## Figures and Tables

**Figure 1 pediatrrep-17-00070-f001:**
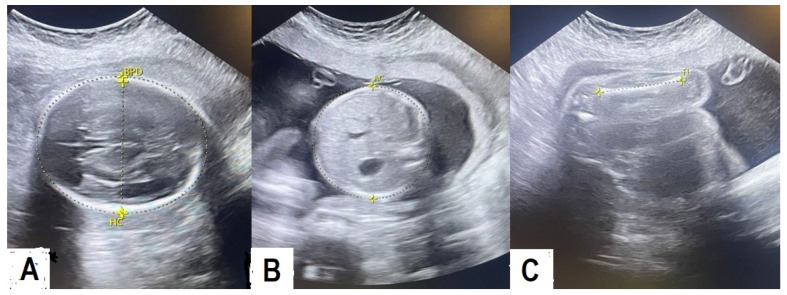
Sonographic parameters used to estimate fetal weight are as follows: (**A**) fetal head circumference (HC) and biparietal diameter (BPD) measured at the level of the thalami, cavum septum pellucidum, and third ventricle; (**B**) abdominal circumference (AC) measured at the level of the umbilical vein, stomach, spine, and posterior rib; (**C**) measurement of the femur length.

**Figure 2 pediatrrep-17-00070-f002:**
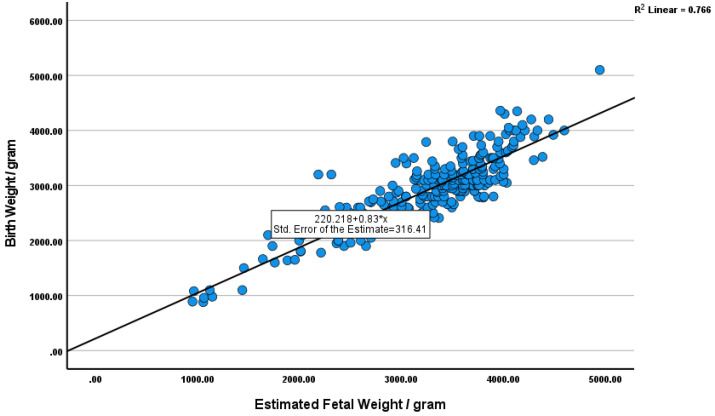
Linear regression analysis showing a strong correlation between EFW and BW (R = 0.875, R^2^ = 0.766, and *p* < 0.001). Dependent variable: birth weight (BW); predictors: (constant) and estimated fetal weight (EFW).

**Figure 3 pediatrrep-17-00070-f003:**
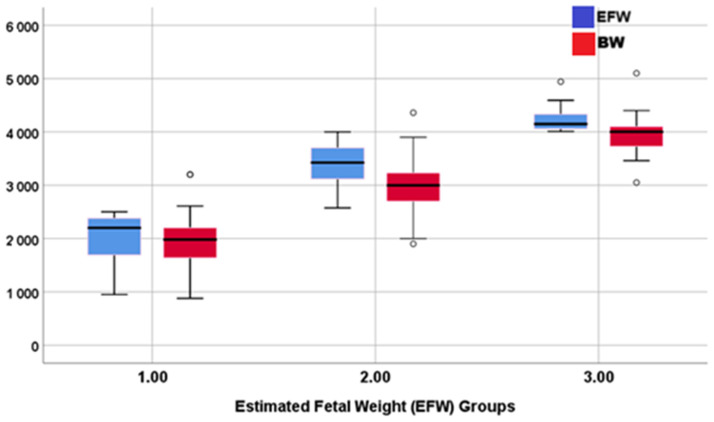
Boxplot comparing the estimated fetal weight (EFW) and birth weight (BW).

**Table 1 pediatrrep-17-00070-t001:** Means of the estimated fetal weight (EFW) and birth weight (BW).

Variables	No.	Minimum	Maximum	Mean ± St. Deviation	Percentiles		
					25th	50th (Median)	75th
Estimated Fetal Weight (EFW)	270	951.00	4942.00	3271.55 ± 691.47	2919.25	3392.50	3745.50
Birth Weight (BW)	270	880.00	5100.00	2918.08 ± 652.79	2587.50	3000.00	3300.00

**Table 2 pediatrrep-17-00070-t002:** Cross-tabulation test showing compatibility between the EFW and BW.

Variables	Birth Weight (BW)				
	Categories	Group-1	Group-2	Group-3	Total
Estimated fetal weight (EFW)	Group-1	33 (86.8%)	5 (13.2%)	0 (0.0%)	38
	Group-2	29 (14.0%)	177 (85.5%)	1 (0.5%)	207
	Group-3	0 (0.0%)	18 (72.0%)	7 (28.0%)	25
	Total	62 (23%)	200 (74.0%)	8 (3%)	270 (100%)

**Table 3 pediatrrep-17-00070-t003:** Non-parametric Mann–Whitney U test to compare BW in both fetal genders.

Gender	n	Mean ± SD	Median	Mean Rank	U	z	*p*-Value	r
Male	145	2945.37 ± 730.83	3000	143.59	7890	−1.83	0.067	0.11
Female	125	2886.43 ± 549.83	2910	126.12				

n: number, SD: standard deviation, U: U test, z: z-value, r: correlation coefficient.

**Table 4 pediatrrep-17-00070-t004:** Prediction of birth weight based upon mother’s BMI, height, and weight.

Model		Unstandardized Coefficients		Standardized Coefficients	t	Sig.	95.0% CI for B	R and (R^2^)
		B	Std. Error	Beta				
1	(Constant)	2253.241	4103.506		0.549	0.583	−5826.24–10,332.73	0.169 and 0.028
	Weight	11.450	26.487	0.250	0.432	0.666	−40.7–63.6	
	Height	1.169	26.167	0.010	0.045	0.964	−50.35–52.69	
	BMI	−11.761	65.435	−0.100	−0.180	0.857	−140.597–117.075	

Dependent variable: BW (birth weight); predictors: (constant), mother’s weight, height, and BMI. B: the beta coefficient, t: the test statistics, CI: confidence interval, BMI: body mass index, BW: birth weight, R: correlation coefficient, R^2^: the coefficient of determination.

## Data Availability

Data are available from the corresponding author upon reasonable request.

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
