# Peer review of "Investigating the Accuracy of Ultrasound Imaging in Measuring Fetal Weight in Comparison with the Actual Postpartum Weight"

_pediatrrep, 2025, doi:10.3390/pediatric17040070_

Round 1
Reviewer 1 Report
Comments and Suggestions for Authors
I have read the paper comparing US measured and real birth weights with great interest. The paper is very interesting and decently presented. I have only minor comments:
1) median interval between EFW by ultrasound examination and birth was reported, but it would also be of interest to see the range.
2) coeficients of variation (SD/mean) could be presented in the text when commenting on wide variation within measured values to obtain better insight into the variability.
3) can authors also present predictors of EFW same as they presented for birth weight?
4) among limitations, authors should mention that sample size may affect statical power of some analyses, especially considering that no relationship with mother's weight was found, which is not completely consistent with published literature.
Author Response
|
Response to Reviewer 1 Comments
|
|
|
|
|
|
Thank you very much for taking the time to review this manuscript. Please find the detailed responses below and the corresponding revisions/corrections highlighted/in track changes in the re-submitted files.
|
|
|
Point-by-point response to Comments and Suggestions for Authors: |
|
|
Comments 1: [median interval between EFW by ultrasound examination and birth was reported, but it would also be of interest to see the range.] |
|
|
Response 1: [The maximum interval was 48 hours] Thank you for pointing this out. I/We agree with this comment. Therefore, I/we have changed the interval between EFW by Ultrasound examination and birth to be maximum of 48 hours. Really, all ultrasound examinations were done within 48 hours before pregnancy. “[Maximum interval between EFW by ultrasound examination and birth was 48 hours.]” |
|
|
Comments 2: [coefficients of variation (SD/mean) could be presented in the text when commenting on wide variation within measured values to obtain better insight into the variability.] |
|
|
Response 2: Agree. I/We have, accordingly, done/revised/changed/modified to emphasize this point. [We add “(3271.55±691.47g vs 2918.08±652.79g)” in the first paragraph (page-6, line-162) of discussion section].
Comments 3: [can authors also present predictors of EFW same as they presented for birth weight?] Response 3 Thank you for pointing this out. The predictors of EFW are the same of the predictors of BW. The EFW is the fetal weight measured by ultrasound, and the BW is the actual fetal weight after birth. The predictors are the same.
Comments 4: [among limitations, authors should mention that sample size may affect statical power of some analyses, especially considering that no relationship with mother's weight was found, which is not completely consistent with published literature.] Response 4: Thank you for pointing this out. I/We agree with this comment. Therefore, I/we have added this in our limitations. We added this paragraph to limitations page-7, lines-2016-219“[Relatively small sample size may affect statical power of some analyses such as in-compatibility between our results and previous studies regarding the correlation be-tween the fetal weight and mother's weight, height, and BMI.]”
|
|
Reviewer 2 Report
Comments and Suggestions for Authors
- This article highlights the correlation between the sonographic fetal biometry with the actual birth weight. Actually, the study is interesting and useful in the field of obstetrics and pediatrics. However, there are some issues in novelty, as by reviewing the literature, there are many publications dealing with this topic. please defend your article novelty.
- The accuracy of ultrasound estimation of fetal weight in comparison to birth weight: A systematic review. https://doi.org/10.1177/1742271X17732807
- Accuracy of immediate antepartum ultrasound estimated fetal weight and its impact on mode of delivery and outcome - a cohort analysis. https://doi.org/10.1186/s12884-018-1772-7
- Please add to the introduction section the research gab
- Please add the hypothesis of this study in the introduction section
- Please add the statistical program you used
Author Response
|
Response to Reviewer 2 Comments
|
||
|
Summary |
|
|
|
Thank you very much for taking the time to review this manuscript. Please find the detailed responses below and the corresponding revisions/corrections highlighted/in track changes in the re-submitted files.
|
||
|
Point-by-point response to Comments and Suggestions for Authors: |
||
|
Comments 1: [please defend your article novelty.] |
||
|
Response 1: Thank you for pointing this out. We have added the following: “[This is the first study exploring this critical topic in our region.]”in the introduction section, page-2, line-66. |
||
|
Comments 2: [Please add to the introduction section the research gab.] |
||
|
Response 2: Thank you for pointing this out. We have added the following: “[In our region, we did not find previous studies covering this significant topic.]”in the introduction section, page-2, line-65.
Comments 3: [Please add the hypothesis of this study in the introduction section.] Response 3 Thank you for pointing this out. We have added the following: “We hypothesized that ultrasonography, when performed by experienced operator, is an accurate imaging method in measuring weight of the fetus” to introduction section, page-2, line 67.
Comments 4: [Please add the statistical program you used.] Response 4: Thank you for pointing this out. The statistical program was already added. We have added the extended name of SPSS in methodology section, page-3, line-102.
|
||
Reviewer 3 Report
Comments and Suggestions for Authors
Dear all, I suggest:
Remove the topics from the abstract.
Include more specific keywords.
Improve the introduction, divide it into at least 5 paragraphs, talk more about the relationship between ultrasound and fetal measurement. Present the main deficiencies and/or problems when performing this evaluation. Add the study hypothesis before the objective.
Add the number of the ethics committee that approved the study.
Add the formulas for each calculation performed, and at the end cite the authors. We cannot keep looking for formulas in other studies, they should already be in the manuscript.
Improve the presentation of tables 1 and 2. I cannot understand them with maximum precision. It might be interesting to put them in Figures or merge the tables.
Figure 2 is not necessary, because the information described before it already says everything about it.
I suggest reorganizing the results as they are, there are many tables... change that, restructure it, add the p-value within the tables for each result presented, this will help the reader.
In the discussion, divide the paragraphs and discuss each result as they were cited, there is no need to include the values.
The conclusion needs to be improved, answer the objective and hypothesis of the study, present a recommendation based on the data you found.
Author Response
|
Response to Reviewer 3 Comments
|
||
|
Summary |
|
|
|
Thank you very much for taking the time to review this manuscript. Please find the detailed responses below and the corresponding revisions/corrections highlighted/in track changes in the re-submitted files.
|
||
|
Point-by-point response to Comments and Suggestions for Authors: |
||
|
Comments 1: [Remove the topics from the abstract.] |
||
|
Response 1: Thank you for pointing this out. Topics were removed from the abstract. Now, the abstract is written in the narrative style. |
||
|
Comments 2: [Include more specific keywords.] |
||
|
Response 2: Thank you for pointing this out. We replaced the keyword “ultrasonography” by the more specific keyword “Antenatal ultrasonography”.]”in the keyword section, page-1, line-32.
Comments 3: [Improve the introduction, divide it into at least 5 paragraphs, talk more about the relationship between ultrasound and fetal measurement. Present the main deficiencies and/or problems when performing this evaluation. Add the study hypothesis before the objective.] Response 3 Thank you for pointing this out. We have divided the introduction into three paragraphs as the following: Paragraph-1: defines ultrasonography and how to measure fetal weight, Paragraph-2: mentioned the significant of antenatal ultrasonography, and Paragraph-3: mentioned the gap, novelty, hypothesis, and significance of the study. Regarding hypothesis: we have added the following: “In our region, we did not find previous studies covering this significant topic. This is the first study exploring this critical topic in our region. We hypothesized that ultra-sonography, when performed by experienced operator, is an accurate imaging method in measuring weight of the fetus.” to introduction section, page-2, lines 66-69.
Comments 4: [Add the number of the ethics committee that approved the study.] Response 4: Thank you for pointing this out. The No. of the IRB was already added in Ethical approval section, page-7, lines-235-237.
Comments 5: [Add the formulas for each calculation performed, and at the end cite the authors. We cannot keep looking for formulas in other studies, they should already be in the manuscript.] Response 5: Thank you for pointing this out. We have added the formulas in methodology section, page-3, line-98-105. Two references were added (References No. 8 and 9), and the order of the next references was changed accordingly.
|
||
Comments 6: [Improve the presentation of tables 1 and 2. I cannot understand them with maximum precision. It might be interesting to put them in Figures or merge the tables.]
Response 6: Thank you for pointing this out. The statistical program was already added. We have added the extended name of SPSS in methodology section, page-3, line-102.
Comments 7: [Figure 2 is not necessary, because the information described before it already says everything about it.]
Response 7: We apology. Table 2 reflects the significant correlation and distribution of the EFW and BW.
Comments 8: [I suggest reorganizing the results as they are, there are many tables... change that, restructure it, add the p-value within the tables for each result presented, this will help the reader.]
Response 8: Table 2 was deleted because its contents are summarized in the text. However, other tables are essential. Table 1 cannot be presented as figure.
Comments 9: [In the discussion, divide the paragraphs and discuss each result as they were cited, there is no need to include the values.]
Response 9: Thank you for pointing this out. We added only the descriptive values of the results, such as the mean±SD, p-value, and r.
Comments 10: [The conclusion needs to be improved, answer the objective and hypothesis of the study, present a recommendation based on the data you found.]
Response 10: Thank you for pointing this out. Modification in conclusion was done.
Reviewer 4 Report
Comments and Suggestions for Authors
In this work, the Authors aimed to investigate the accuracy of ultrasonography to measure the EFW in comparison with the actual birth weight (BW) measured immediately after delivery. A structured data sheet was used to record the estimated fetal weight (EFW), calculated using the Hadlock A formula via real-time ultrasound imaging performed on the day of delivery or the preceding day, as well as the actual birth weight (BW) measured immediately after delivery. They found a strong compatibility between ultrasonography EFW and BW measurements of fetal weight, allowing to achieve high clinical value within the obstetric assessment and subsequently management.
The work presents some discrepancies that need to be addressed through corrections and updates.
Lines 41-43: A better description of the algorithm designed by Hadlock could help a better understanding of the aim of the work.
Line 68: Report also in this section the ethical approval number.
Lines 161-163: Which are the main results reported by Tawe et al.? Describe the correlation between their results and the ones of this study.
Lines 164-171: See comment above. A better comparison among the study results is required.
Lines 189-191: Which could be the reason?
Lines 199-202: How could these discrepancies be explained?
Author Response
|
Response to Reviewer 4 Comments |
||
|
Summary |
|
|
|
Thank you very much for taking the time to review this manuscript. Please find the detailed responses below and the corresponding revisions/corrections highlighted/in track changes in the re-submitted files.
|
||
|
Point-by-point response to Comments and Suggestions for Authors: |
||
|
Comments 1: [Lines 41-43: A better description of the algorithm designed by Hadlock could help a better understanding of the aim of the work.] |
||
|
Response 1: Thank you for pointing this out. I/We agree with this comment. Full description of Hadlock formula and formulas used to measure the EFW was added in the methodology section, page-3, lines-98-105. |
||
|
Comments 2: [Line 68: Report also in this section the ethical approval number.] |
||
|
Response 2: Agree. IRB No. was added, page-2, line-72.
Comments 3: Lines 161-163: Which are the main results reported by Tawe et al.? Describe the correlation between their results and the ones of this study.] Response 3 Thank you for pointing this out. This correlation was clarified, page-7, line-175-176.
Comments 4: [Lines 164-171: See comment above. A better comparison among the study results is required.] Response 4: Thank you for pointing this out. This was declared in page-7, lines-178-179.
Comments 5: [Lines 189-191: Which could be the reason?] Response 5: Thank you for pointing this out. The reason may be overestimation in measurements. |
||
Round 2
Reviewer 2 Report
Comments and Suggestions for Authors
Thanks for responses raised by the authors